

# Prioritizing bona fide bacterial small RNAs with machine learning classifiers

Erik J.J. Eppenhof[1] and Lourdes Peña-Castillo[2,3]

[1] Department of Artificial Intelligence, Radboud University Nijmegen, Nijmegen, Netherlands
[2] Department of Biology, Memorial University of Newfoundland, St. John's, Canada
[3] Department of Computer Science, Memorial University of Newfoundland, St. John's, Canada

## ABSTRACT

Bacterial small (sRNAs) are involved in the control of several cellular processes. Hundreds of putative sRNAs have been identified in many bacterial species through RNA sequencing. The existence of putative sRNAs is usually validated by Northern blot analysis. However, the large amount of novel putative sRNAs reported in the literature makes it impractical to validate each of them in the wet lab. In this work, we applied five machine learning approaches to construct twenty models to discriminate bona fide sRNAs from random genomic sequences in five bacterial species. Sequences were represented using seven features including free energy of their predicted secondary structure, their distances to the closest predicted promoter site and Rho-independent terminator, and their distance to the closest open reading frames (ORFs). To automatically calculate these features, we developed an sRNA Characterization Pipeline (sRNACharP). All seven features used in the classification task contributed positively to the performance of the predictive models. The best performing model obtained a median precision of 100% at 10% recall and of 64% at 40% recall across all five bacterial species, and it outperformed previous published approaches on two benchmark datasets in terms of precision and recall. Our results indicate that even though there is limited sRNA sequence conservation across different bacterial species, there are intrinsic features in the genomic context of sRNAs that are conserved across taxa. We show that these features are utilized by machine learning approaches to learn a species-independent model to prioritize bona fide bacterial sRNAs.

## INTRODUCTION

Bacterial small RNAs (sRNAs) are ubiquitous regulators of gene expression, mostly acting by antisense mechanisms on multiple target mRNAs and, as a result of this, they are involved in the control of many processes such as adaptive responses, stress responses, virulence, and pathogenicity (*Storz, Vogel & Wassarman, 2011*; *Michaux et al., 2014*). Numerous (hundreds) putative sRNAs have been identified in many bacterial species through RNA sequencing (RNA-seq) (e.g., *Grüll et al. (2017)*; *Thomason et al. (2015)*; *Zeng & Sundin (2014)*; *McClure, Tjaden & Genco (2014)*). The existence of putative sRNAs is usually validated by Northern blot analysis. However, the large amount of novel putative sRNAs

Corresponding author
Lourdes Peña-Castillo,
lourdes@mun.ca

reported in the literature makes it impractical to validate in the wet lab each of them. To optimize resources, one would like to first investigate those putative sRNAs which are more likely to be bona fide sRNAs. To do that, we need to computationally prioritize sRNAs based on their likelihood of being bona fide sRNAs. Despite the fact that tools to tackle this problem have been around since early 2000s (*Lu, Goodrich-Blair & Tjaden, 2011*; *Backofen & Hess, 2010*), computational prediction of sRNAs in genomic sequences remains a challenging unsolved problem. Available tools typically use comparative genomics, primary sequence and secondary structure features to predict whether a genomic sequence corresponds to an sRNA, with the comparative genomics-based prediction of sRNAs being the standard method (*Lu, Goodrich-Blair & Tjaden, 2011*; *Backofen & Hess, 2010*). As there are many species-specific sRNAs, and functionally equivalent sRNAs show very low sequence conservation (*Wagner & Romby, 2015*), a comparative genomics-based model for sRNA prediction (as used by most tools) is not suitable for the majority of sRNAs and many sRNAs are excluded from these predictions. Additionally, as very limited overlap between sRNAs detected by RNA-seq and sRNAs predicted by bioinformatic tools has been observed in several studies (*Soutourina et al., 2013*; *Wilms et al., 2012*; *Vockenhuber et al., 2011*), available tools are not suitable for quantifying the probability of a putative sRNA detected from RNA-seq data being indeed a genuine sRNA. A machine learning-based approach using genomic context features for sRNA prioritization may be able to overcome the issues caused by the limited sequence conservation of most sRNAs, and to detect intrinsic features of sRNA sequences common to a number of bacterial species.

The main goal of this study was to develop a bioinformatic tool (applicable to a wide range of bacterial species) to allow microbiologists to prioritize sRNAs detected from RNA-seq data based on their probability of being bona fide sRNAs. To do this, we comparatively assessed the performance of five machine learning approaches for quantifying the probability of a genomic sequence encoding a bona fide sRNA. The machine learning approaches applied were: logistic regression (LR), multilayer perceptron (MP), random forest (RF), adaptive boosting (AB) and gradient boosting (GB). We used data from five bacterial species including representatives from the phyla *Firmicutes* (*Streptococcus pyogenes*), *Actinobacteria* (*Mycobacterium tuberculosis*), and *Proteobacteria* (*Escherichia coli*, *Salmonella enterica*, and *Rhodobacter capsulatus*). To assess the applicability of the methods to a wide range of bacteria, we evaluated the methods in bacterial species not included in the training data. As input to the machine learning approaches, we provided a vector of seven features per sequence. These features are: the free energy of the predicted secondary structure, distance to their closest predicted promoter site, distance to their closest predicted Rho-independent terminator, distances to their two closest open reading frames (ORFs), and whether or not the sRNA is transcribed on the same strand as their two closest ORFs. These features were selected under the hypothesis that genomic context and secondary structure of sRNAs are better preserved across diverse bacteria than sequence characteristics such as frequencies of mono-nucleotides, di-nucleotides, and tri-nucleotides. This hypothesis is based on the fact that there is low sequence conservation of sRNAs among bacteria (*Wagner & Romby, 2015*), and our observation that genomic context of putative sRNAs is distinct from that of random genomic sequences (*Grüll et al., 2017*). We tested our

best performing model in a multi-species dataset (*Lu, Goodrich-Blair & Tjaden, 2011*) and the performance achieved by our method (sRNARanking) demonstrated that it is possible to create a highly accurate and general (i.e., species-independent) model for prioritizing bona fide bacterial sRNAs using genomic context features.

Obtaining the selected sRNA features requires the use of numerous different bioinformatic tools which may be challenging for the average user. To facilitate sRNA characterization, we have developed sRNACharP (**sRNA Char**acterization **P**ipeline), a pipeline to automatically compute the seven features used by sRNARanking (available at https://github.com/BioinformaticsLabAtMUN/sRNACharP). To enable other researchers to use sRNARanking, we made an R script available containing the model (https://github.com/BioinformaticsLabAtMUN/sRNARanking). We expect that together these two tools (sRNACharP and sRNARanking) will facilitate and accelerate the characterization and prioritization of putative sRNAs helping researchers in the field of RNA-based regulation in bacteria to focus in the putative sRNAs most likely to be bona fide sRNAs.

# METHODS

## Datasets

Published positive instances of bona fide sRNAs were collected for *R. capsulatus* (*Grüll et al., 2017*), *M. tuberculosis* (*Miotto et al., 2012*), *S. pyogenes* (*Le Rhun et al., 2016*), and *S. enterica* (*Kröger et al., 2012*). *M. tuberculosis*, *S. pyogenes* and *S. enterica* positive instances have all been verified by Northern blot analysis; while, *R. capsulatus* positive instances included, in addition to four experimentally verified sRNAs, 41 homologous sRNAs (i.e., sRNAs that have high sequence similarity to known sRNAs in other bacterial species or were found to be conserved in the genome of at least two other bacterial species). Additionally, we collected *E. coli* sRNAs, supported by literature with experimental evidence from RegulonDB (release 9.3) (*Gama-Castro et al., 2016*).

To build our models we randomly selected 80% of the bona fide sRNAs of *R. capsulatus*, *S. pyogenes* and *S. enterica*. To estimate false positive predictions and build our models, for each bacterial species we created a set of negative instances by generating random genomic sequences that do not overlap with the positives instances for the particular bacterium. Basically, negative instances are sets of randomly selected genomic regions where there is no experimental evidence for the presence of sRNAs. Negative instances match the length and the strand of the positive instances. To generate the negative instances, we used BEDTools (*Quinlan & Hall, 2010*) (code available in Supplemental File). We randomly selected $n$ negative instances for training, where $n$ is three times the number of positive instances in the corresponding training set. In previous similar studies, $n$ has been set to be one (*Arnedo et al., 2014*) or two (*Barman, Mukhopadhyay & Das, 2017*) times the number of positive instances. However, we believed that a more unbalanced dataset for training is closer to a real scenario, and decided to increase the value of $n$ to three times the number of positive instances. All remaining negative instances were used for testing the models.

An alternative approach to generate negative instances is to take a genome sequence and randomly shuffle the order of its bases as done by *Arnedo et al. (2014)* and *Barman,*

**Table 1  The number of positive (bona-fide sRNAs) and negative (random genomic sequences) instances in the datasets used for training and testing the classification models.** The NCBI accession number of the genome sequence used is indicated in the first column between brackets. The "Combined" data are made by putting together the training data of *S. enterica*, *S. pyogenes* and *R. capsulatus*.

| | Training | | Test | |
|---|---|---|---|---|
| | Positive instances | Negative instances | Positive instances | Negative instances |
| *R. capsulatus* (NC_014034.1) | 36 | 108 | 9 | 342 |
| *S. pyogenes* (NC_002737.2) | 37 | 110 | 9 | 349 |
| *S. enterica* (NC_016810.1) | 90 | 271 | 23 | 855 |
| Combined | 163 | 489 | N/A | N/A |
| *E. coli* (NC_000913.3) | N/A | N/A | 125 | 1245 |
| *M. tuberculosis* (NC_000962.3) | N/A | N/A | 19 | 190 |

*Mukhopadhyay & Das (2017)*. However, as we use genomic context features for representing the genomic sequences, shuffling the sequences' bases would preserve their genomic context properties and therefore be ineffective. Furthermore, as mentioned by *Arnedo et al. (2014)* and *Lu, Goodrich-Blair & Tjaden (2011)*, the use of non-annotated genomic sequences as negative instances gives a more conservative estimate of the precision of the models.

The number of positive and negative instances per bacterium used for training and testing the machine learning models is shown in Table 1. Training and test data are provided in Supplemental File.

## sRNA Characterization

Each sRNA is represented as a vector of seven numerical features or attributes, as in *Grüll et al. (2017)*. These attributes are:

1. free energy of the sRNA predicted secondary structure,
2. distance to the closest -10 promoter site predicted in the genomic region starting 150 nts upstream of the start of the sRNA sequence to the end of the sRNA sequence (if no promoter site is predicted in that region a value of $-1,000$ is used),
3. distance to the closest predicted Rho-independent terminator in the range of $[0,1,000]$ nts (if no terminator is predicted within this distance range a value of 1,000 is used),
4. distance to the closest left ORF, which is in the range of $(-\infty, 0]$ nts,
5. a Boolean value (0 or 1) indicating whether the sRNA is transcribed on the same strand as its left ORF,
6. distance to the closest right ORF, which is in the range of $[0, +\infty)$, and
7. a Boolean value indicating whether the sRNA is transcribed on the same strand as its right ORF.

A "left" ORF is an annotated ORF located at the 5′ end of a genomic sequence on the forward strand or located at the 3′ end of a genomic sequence on the reverse strand (Fig. 1). A "right" ORF is an annotated ORF located at the 3′ end of a genomic sequence on the forward strand or located at the 5′ end of a genomic sequence on the reverse strand.

To automatically calculate these seven features for a set of sRNAs from a given bacterial species, we developed sRNACharP. As input, sRNACharP requires only a BED file
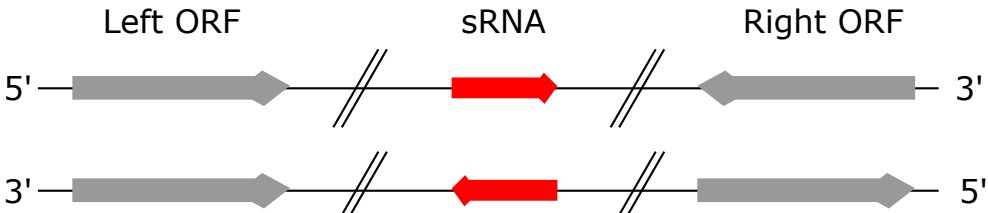

**Figure 1** **Left and right ORFs.** Left ORFs are located at the 5′ end of an sRNA on the forward strand or at the 3′ end of an sRNA on the reverse strand. Right ORFs are located at the 3′ end of an sRNA on the forward strand or at the 5′ end of an sRNA on the reverse strand.

(*UCSC website, 2018*) with the genomic coordinates of the sRNAs, a FASTA file with the corresponding genome sequence, and a BED file with the genomic coordinates of the annotated protein coding genes (ORFs). sRNACharP is implemented in Nextflow (*Di Tommaso et al., 2017*) and available at https://github.com/BioinformaticsLabAtMUN/ sRNACharP. To ensure reproducible results and reduce installation requirements to the minimum, sRNACharP is distributed with a Docker container (*Di Tommaso et al., 2015*). sRNACharP uses the following bioinformatic tool (the versions listed within brackets are the ones installed in the Docker container). CentroidFold (*Hamada et al., 2009*) (version 0.0.15) with parameters `-e ''CONTRAfold''` and `-g 4` is used to predict the secondary structure of the sequences given. BEDtools' slopBed and fastaFromBed (*Quinlan & Hall, 2010*) (version 2.26) are used to extract the sRNA sequences, and the sequences including 150 nts upstream of the 5′ end of the sRNAs in FASTA format. Promoter sites on the sequences including 150 nts upstream of the 5′ end of the sRNAs are predicted using BPROM (*Solovyev & Salamov, 2011*) with default values. Rho-independent terminators are predicted using TransTermHP (*Kingsford, Ayanbule & Salzberg, 2007*) (version 2.09) with default values. Alternatively, sRNACharP can take as input, files from the TransTermHP website (http://transterm.cbcb.umd.edu/cgi-bin/transterm/predictions.pl). For this study, we downloaded the predicted Rho-independent terminators for *S. pyogenes* and *M. tuberculosis* from the TransTermHP website on March 2017. The distances to the closest terminator and the closest ORFs are obtained using BEDtools' closest. Finally, R (version 3.4.4) is used to generate the features table.

## Machine learning approaches

We assessed the performance of logistic regression (*Cox, 1958*; *Walker & Duncan, 1967*), multilayer perceptron (*Bishop, 1995*; *Fahlman, 1988*), random forest (*Breiman, 2001*) and boosting models (*Schapire, 1990*) for the task of quantifying the probability of a genomic sequence encoding a bona fide sRNA. Random forests and boosting classifiers are both examples of ensemble learning algorithms (*Dietterich, 2000*). The core of the boosting methods lies in iteratively combining outputs of so-called "weak learners", converging to an overall strong learner. Logistic regression (LR) was used in *Grüll et al. (2017)* and showed to outperform linear discriminant analysis (LDA) and quadratic discriminant analysis (QDA) for this task. We decided to use LR as a baseline to compare the performance of the other

classifiers. We chose to compare the other four machine learning approaches (classifiers) because they have shown to perform well on small datasets and they are generally robust to noise (*Liaw & Wiener, 2002*; *Kerlirzin & Vallet, 1993*; *Ridgeway, 1999*).

All the machine learning classification approaches were implemented in the Python programming language version 3.6. Scikit-learn (version 0.19.1) (*Pedregosa et al., 2011*) was used for the implementation of all the classifiers. For each classifier, the "best" parameters were obtained by maximizing the average area under the ROC curve (AUC) when performing leave-one-out cross-validation (LOO CV) on the training data (Fig. 2). The final "best" parameters per classifier used were those given on average the largest AUC.

### Logistic regression

Logistic Regression (LR) learns the parameters $\beta$ of the logistic function,

$$p(X) = \frac{e^{\beta_0 + \beta_1 X_1 + \ldots + \beta_n X_n}}{1 + e^{\beta_0 + \beta_1 X_1 + \ldots + \beta_n X_n}},$$

where $p(X)$ is the probability of an sRNA with feature vector $X$ of being a bona fide sRNA, $e$ is the base of the natural logarithm, $n$ is the number of features, and $X_i$ is the value of feature $i$. To fit the model, usually the maximum likelihood approach is used. We used the balanced mode that automatically adjust class weights inversely proportional to class frequencies in the input data. All other parameters were left to their default values.

### Multilayer perceptron

Multilayer Perceptrons (MPs) are fully connected feed-forward neural networks, with one or more layers of hidden nodes between the input and output nodes (*Bishop, 1995*; *Fahlman, 1988*). Except for the input node(s), each node is a neurone with a nonlinear activation function. Each neurone combines weighted inputs by computing their sum to determine its output based on a certain threshold value and the activation function. The output $y$ of the system can be described as

$$y = f\left(\sum_{i=0}^{N} w_i x_i\right),$$

where $x_1, \ldots, x_N$ represent the input signals, $w_1, \ldots, w_N$ are the synaptic weights and $f$ is the activation function. MPs learn through an iterative process of changing connection weights after processing each part of the data. The most common learning algorithm used for this process is backpropagation (*Fahlman, 1988*).

The activation function that lead to the largest AUCs on the training data was the logistic sigmoid function. We used the standard backpropagation algorithm with an initial random generation of weights ($[-1, 1]$). As using multiple hidden layers decreased the performance, we decided to use only one hidden layer. The number of hidden nodes explored was in the range from 1 (in that case the model behaves the same as logistic regression) to 1,000 with steps of 50. The best number of hidden nodes was found to be 400. Learning rates ranging from 0.1 to 1.0 were explored in steps of 0.1. The chosen learning rate was a constant learning rate of 0.9, because an adaptive learning rate was observed to decrease AUCs. The L2 penalty was set to the default value of 0.0001.
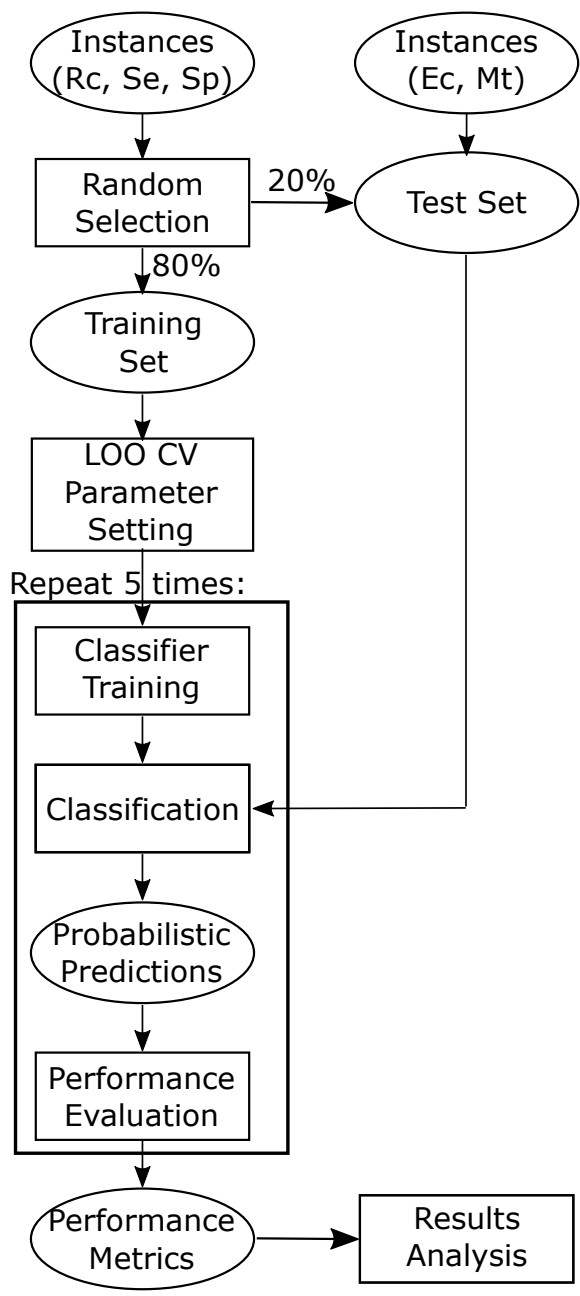

**Figure 2 Flowchart depicting training and testing methodology.** Training and test datasets are labelled with the corresponding bacterial species: Ec, *Escherichia coli*, Mt, *Mycobacterium tuberculosis*, Se, *Salmonella enterica*, Sp, *Streptococcus pyogenes*, and Rc, *Rhodobacter capsulatus*.

### Random forest

A random forest (RF) is constructed by combining multiple decision trees during training (*Breiman, 2001*). All decision trees in the random forest contribute to the determination of the final output class. The output class is determined by averaging the probabilities produced by the individual trees. The range of number of estimators (decision trees)

explored was from 1 to 1,000 in steps of 100. The best setting was found to be 400. The largest AUC results were obtained when the nodes are expanded until almost all leaves are pure. We tested our model with the maximum depth of the tree ranging from 15 to 25 and found that the maximum AUC was obtained at a depth of 20. All features were used in every tree. To measure the quality of a split we used the default Gini index (*Strobl, Boulesteix & Augustin, 2007*) and the maximum number of features to consider when looking for the best split in a node was set to 2, as calculated by the function tuneRF available in the R package randomForest (version 4.6–12).

### Adaptive boosting

Adaptive boosting or AdaBoost (AB) was developed for binary classification problems and tweaks "weak learners" by focusing on the instances that were wrongly classified by previous classifiers (*Freund & Schapire, 1997*). Therefore the training error decreases over the iterations. The additive model of AdaBoost can be formulated as following. The output of each weak learner is described by:

$$L_K(x) = \sum_{k=1}^{K} l_k(x).$$

where $K$ is the total number of iterations and $l_k(x)$ is the output function of the weak learner when taking the instance $x$ as input. To minimize the training error $E_k$ for each iteration $k$, AdaBoost uses:

$$E_k = \sum_{i=1}^{N} E(L_{k-1}(x_i) + \alpha_k h(x_i)),$$

where $h(x_i)$ is the predicted output of a weak learner for every instance $x_i$ in the training set, $\alpha_k$ is the assigned coefficient that minimizes the training error, and $N$ is the total number of instances in the training set.

We used AdaBoost on a random forest (RF) classifier that performed just better than chance on the training data. The best parameters of this RF were found to be 100 decision trees (estimators) and a maximum depth of 1. This means all of the trees were decision stumps. The number of estimators was established at 100 after exploring a range from 1 to 1,000 estimators with steps of 50. A maximum depth of 1 was chosen because AdaBoost is known to perform better with decision stumps (*Ridgeway, 1999*).

### Gradient boosting

In gradient boosting (GB) an initial poor fit on the data is improved by fitting base-learners (e.g., decision trees) to the negative gradient of a specified loss function (*Friedman, 2001*). Gradient boosting can be described by:

$$\hat{f} = argmin_f E_{x,y}[\rho(Y, f(X))],$$

where $X = \{x_1, \ldots, x_n\}$ and $Y = \{y_1, \ldots, y_n\}$, forming the training set $\{(x_1, y_1), \ldots, (x_n, y_n)\}$. $\hat{f}$ minimizes expectation $E$ of the loss function $\rho$ over all prediction functions $f$ that take $X$ as input.

We used gradient boosting on 50 estimators (decision trees) with a maximum depth of 15. We established the number of estimators by exploring a range of 1 to 1,000 estimators with steps of 50. We tested our model with the same maximum depth of the tree as for the decision tree classifiers. We then gradually decreased the maximum depth taking steps of 1, arriving at 15 as the best setting. The minimum number of samples at a leaf node was set to 5, as this was the number found to maximize AUC. Stochastic gradient boosting was performed with a subsampling of 0.9.

## Model building

As shown in Fig. 2 we randomly selected 80% of the positive instances for training, while setting aside the other 20% for testing the models. The test sets were held-out sets used to obtain an unbiased estimate of the models performance and were not used to build or fine-tune the models. As the classifiers used construct models stochastically, five training runs were carried out for each of the 20 models (five machine learning approaches times four training sets) to estimate the stability of the models (script available in Supplemental File).

## Performance assessment

Model performance was assessed in terms of the Area Under the Precision-Recall Curve (AUPRC). There are many classification metrics and none of them can reflect all the strengths and weakness of a classifier (*Lever, Krzywinski & Altman, 2016*). We have chosen AUPRC as the PRC shows precision values for corresponding sensitivity (recall) values and is considered more informative than the ROC when evaluating performance on unbalanced datasets (*Saito & Rehmsmeier, 2015*). Additionally, precision is a more relevant measure to many end users, since it represents the proportion of validation experiments for predictions that would prove successful. We used in-house Python and R scripts to calculate the performance metrics and generate plots (code is provided in Supplemental File). In Python we used the functions available in Scikit-learn. In R, we used the packages ROCR (*Sing et al., 2005*) and PRROC (*Grau, Grosse & Keilwagen, 2015*).

Models were evaluated on five test sets. Each test set corresponds to data from one bacterial species. Data of *R. capsulatus*, *S. pyogenes* and *S. enterica* were also used for training, while *E. coli* and *M. tuberculosis* data were used exclusively for testing the models (Table 1). The species not included in the training data were chosen to be one species of the same taxa as and one of a different taxa from the species used for training. Median, mean and standard deviation of the performance measurements across the five training runs were calculated.

Additionally, to highlight the difference in performance between the models, we used a "winner-gets-all" comparison by ranking the methods based on their mean AUPRC for each test set. The model(s) with the highest mean AUPRC for a specific test set were ranked 1 for that test set. Ties were all given the same rank. At the end of the ranking process, each model has five ranks corresponding to one rank per test set.

Analysis of variance (ANOVA) was performed to explore the effects of classifier and training data on the AUPRC values, and the Tukey's Honest Significant Difference (HSD) (*Tukey, 1949*) method was used to assess the significance on the differences between the

mean AUPRC of classifiers, training data, and models. Additionally, statistical significance of the difference in performance between models was estimated using the Friedman test which is a non-parametric test recommended for comparison of more than two classifiers over multiple datasets (*Demšar, 2006*). To find out which models differ in terms of performance, we used the Nemenyi post-hoc test (*Demšar, 2006*), the Quade post-hoc test (*García et al., 2010*) and the Conover post-hoc test (*Conover, 1999*). We used several post-hoc tests as it is recommended to use several comparison tests (*García et al., 2010*). Multiple testing correction was performed using the Benjamini and Hochberg's FDR method (*Benjamini & Hochberg, 1995*) implemented in the R function p.adjust. All statistical analyses were carried out in R using the packages PMCMR (*Pohlert, 2014*) and scmamp (*Calvo & Santafe, 2016*).

## Attribute importance

To gain insight on how important each attribute is in inferring whether or not a sequence encodes a bona fide sRNA, we used the function varImp available in the R package randomForest (version 4.6-12). To use this function, we first created a RF classifier using the randomForest function with ntree set to 400 and mtry set to 2. These were the best parameters found when tuning the RF classifier (see above). We generated the RF model using the combined training data (Table 1). Attribute importance was measured in terms of the mean decrease in accuracy caused by an attribute during the out of bag error calculation phase of the RF algorithm (*Breiman, 2001*). The more the accuracy of the RF model decreases due to the exclusion (or permutation) of a single attribute, the more important that attribute is deemed for classifying the data.

## Assessing the performance of the best model on benchmark datasets

We compared the performance of our best performing model with that of other four existing approaches as estimated previously by *Lu, Goodrich-Blair & Tjaden (2011)* in a multi-species dataset. Additionally, we assessed the performance of Barman et al.'s SVM method (*Barman, Mukhopadhyay & Das, 2017*) on this dataset. We also compared the performance of our best model in a *Salmonella enterica serovar* Typhimurium LT2 (SLT2) dataset with the performance estimated by *Arnedo et al. (2014)* and *Barman, Mukhopadhyay & Das (2017)*. Since Arnedo et al. and Barman et al. used shuffled genome sequences to generate the negative examples, while we are using sequences from random genomic locations, we generated predictions with Barman et al.'s SVM method on exactly our same SLT2 test set. To generate predictions with Barman et al.'s SVM method, we used the R code and proposed best model provided by Barman et al. to calculate the input features and obtain the predictions.

We obtained a table with start position, end position, strand and genome of sRNAs in the multi-species dataset from *Lu, Goodrich-Blair & Tjaden (2011)*'s Table S1. We found that 34 sRNAs in Lu et al.'s dataset were duplicated entries. After removing the duplicated entries we used 754 sRNAs of fourteen different bacterial species (Table 2). We obtained a table with start position, end position, and name of 182 sRNAs in the SLT2 dataset from *Barman, Mukhopadhyay & Das (2017)*'s Table S6. We noticed that 106 out of the

**Table 2  Number of positive instances per bacterial species in Lu et al.'s dataset used in this study.** The NCBI accession number of the genome sequence used is indicated in the first column between brackets.

| Bacterium | Positive instances |
|---|---|
| *Burkholderia cenocepacia* AU 1054 (NC_008060, NC_008061.1, NC_008062.1) | 18 |
| *Bacillus subtilis subsp. subtilis* str. 168 (NC_000964.3) | 12 |
| *Caulobacter crescentus* CB15 (NC_002696.2) | 7 |
| *Chlamydia trachomatis* L2b/UCH-1/proctitis (NC_010280.2) | 23 |
| *Escherichia coli* K12 MG1655 (NC_000913.3) | 79 |
| *Helicobacter pylori* 26695 (NC_000915.1) | 50 |
| *Listeria monocytogenes* EGD-e (NC_003210.1) | 56 |
| *Pseudomonas aeruginosa* PA01 (NC_002516.2) | 17 |
| *Staphylococcus aureus subsp. aureus* N315 (NC_002745.2) | 9 |
| *Streptomyces coelicolor* A3(2) (NC_003888.3) | 3 |
| *Salmonella enterica subsp. enterica serovar* Typhimurium str. LT2 (NC_003197.2) | 115 |
| *Shewanella oneidensis* MR-1 (NC_004347.2) | 9 |
| *Vibrio cholerae* O1 biovar El Tor str. N16961 (NC_002505.1, NC_002506.1) | 137 |
| *Xenorhabdus nematophila* ATCC 19061 (NC_014228.1) | 219 |

182 sRNAs in the SLT2 dataset were also contained in Lu et al.'s dataset, and thus the benchmark datasets are not completely independent. We retrieved the complete genome sequence and genome annotation of the corresponding bacterium from NCBI. We then extracted the corresponding sRNAs sequences using BEDtools, and obtained the feature vectors using our sRNACharP pipeline. We generated the negative instances for each dataset as described above. For Lu et al.'s (2011) dataset, we generated three negative instances for each positive instance; however, the ratio of positive to negative instances used by Lu et al. is not reported in their article and thus the ratio they used may differ from this. For the SLT2 dataset we generated ten negative instances for each positive instance to match the ratio of positive to negative instances used by *Barman, Mukhopadhyay & Das (2017)*. The shuffled negative instances used by Barman et al. were obtained from *Barman, Mukhopadhyay & Das (2017)*'s Tables S7 to S16.

# RESULTS

## Performance assessment

In this section models are identified by the classifier and the training data used. Training and test datasets are labelled with the corresponding bacterial species: Ec, *Escherichia coli*; Mt, *Mycobacterium tuberculosis*; Se, *Salmonella enterica*; Sp, *Streptococcus pyogenes*; and Rc, *Rhodobacter capsulatus*. For each classifier, four models were generated (one for each training set). Each of these four models was evaluated on each of the five test sets. The distribution of AUPRC of all models obtained using each classifier is shown in Fig. 3. As LR

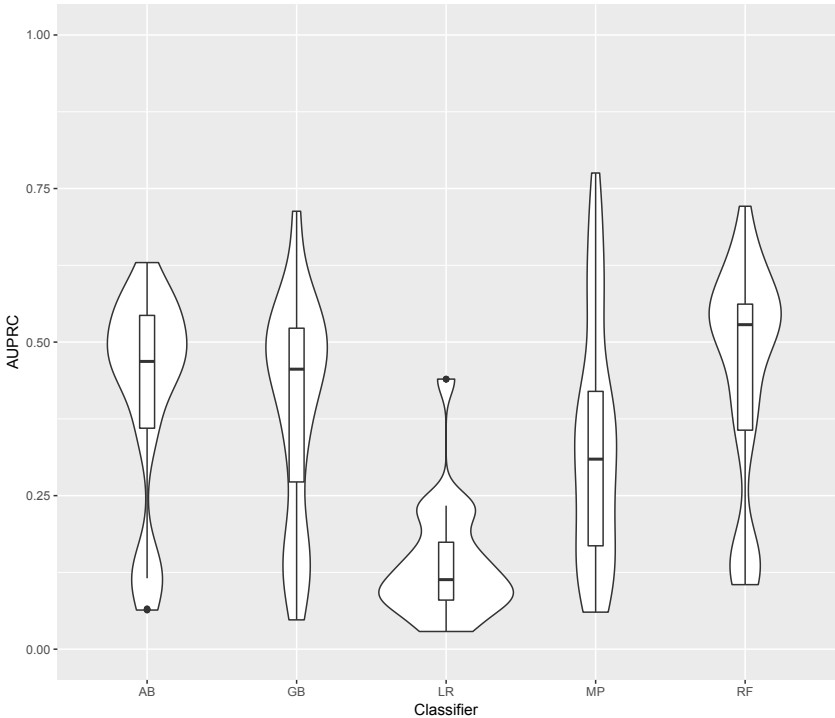

**Figure 3** **Distribution of AUPRC values per classifier.** Violin plots illustrate the distribution of AUPRC values for all models obtained with each classifier. Inside the distribution shape a box indicates the range from the 25 percentile to 75 percentile of the precision values. AB, adaptive boosting, GB, gradient boosting; LR, logistic regression; MP, multilayer perceptron; RF, random forest.

was clearly outperformed by the other four classifiers, we excluded LR results from further analysis.

The mean AUPRC for each of the 16 models is depicted in Fig. 4. For the four classifiers, models trained on the Rc training data have lower AUPRC values than models trained on the other three training sets. The classifier producing the most variable models was MP with average standard deviations above the overall mean standard deviation; while AB was the classifier with the lowest standard deviation. AB is the classifier least susceptible to variations in AUPRC due to the training data; while, MP is the classifier with more variation in AUPRC due to the training data (Fig. 4). ANOVA results indicated that the classifier and the training data are both significant factors to explain variance in AUPRC values ($F$-statistic $= 8.29$, $p$-value $2.34e^{-5}$ and $F$-statistic $= 8.26$, $p$-value $2.46e^{-5}$, respectively). There was not significant interaction between these two factors found by ANOVA.

To emphasize differences in performance among the models, we ranked each model based on mean AUPRC obtained on each test set (see 'Methods'). The model with the highest AUPRC is ranked one, and ties are assigned the same rank. Figure 5 depicts the mean rank of each classifier. The Friedman test ($p$-value $= 0.008$) indicated that the average rank obtained by some of the classifiers is significantly different from the mean rank expected under the null hypothesis. We then used three post-hoc tests for pairwise
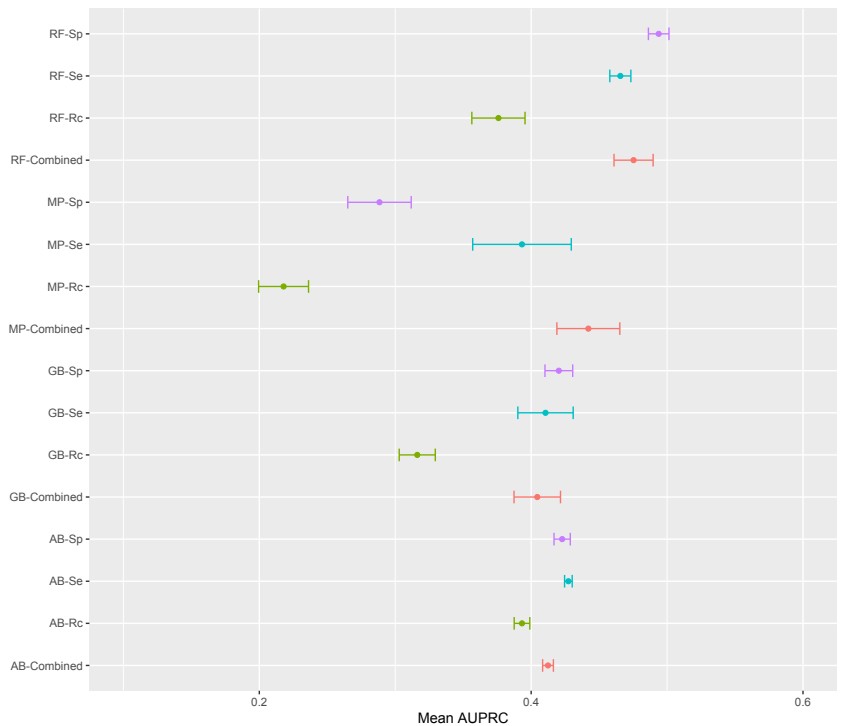

**Figure 4  Mean AUPRC per model.** The dot represents the mean AUPRC and bars represent standard error. Colour indicates the training data used: Red, Combined data, Green, *R. capsulatus* data; Blue, *S. enterica* data; Purple, *S. pyogenes* data. Classifiers are indicated by AB, adaptive boosting; GB, gradient boosting; MP, multilayer perceptron; RF, random forest.

comparisons. The Nemenyi test identified two groups of classifiers with similar ranks: RF, AB and GB in one group, and AB, GB and MP in the other group. According to the Nemenyi test, the ranks of RF are statistically significantly lower ($p$-value $= 0.008$) than the ranks of MP (Fig. 6). The Quade post-hoc test deemed the differences in ranks between RF and MP (FDR corrected $p$-value $= 0.002$) and between RF and GB (FDR corrected $p$-value $= 0.006$) as statistically significant. Finally, the Conover post-hoc test found statistically significant differences between the ranks of RF vs MP (FDR corrected $p$-value $= 0.0005$), RF vs GB (FDR corrected $p$-value $= 0.002$), RF vs AB (FDR corrected $p$-value $= 0.018$), and AB vs MP (FDR corrected $p$-value $= 0.031$). Based on these results, we concluded that RF is the only classifier with significant differences in ranks with respect to the other classifiers. We selected the RF—Combined model as our best performing model, as it has one of the highest mean AUPRC, lowest mean rank and low standard deviation. However, these three models: RF-Combined, RF-Sp and RF-Se are likely comparable in terms of mean rank and AUPRC (Fig. 4 and Fig. S1).

To facilitate other researchers to rank their own sRNAs, we have created sRNARanking, an R script that produces the probabilistic predictions generated by the RF-Combined model. We used R to distribute the model because R is more commonly used by natural science researchers than Python. sRNARanking takes as input the feature
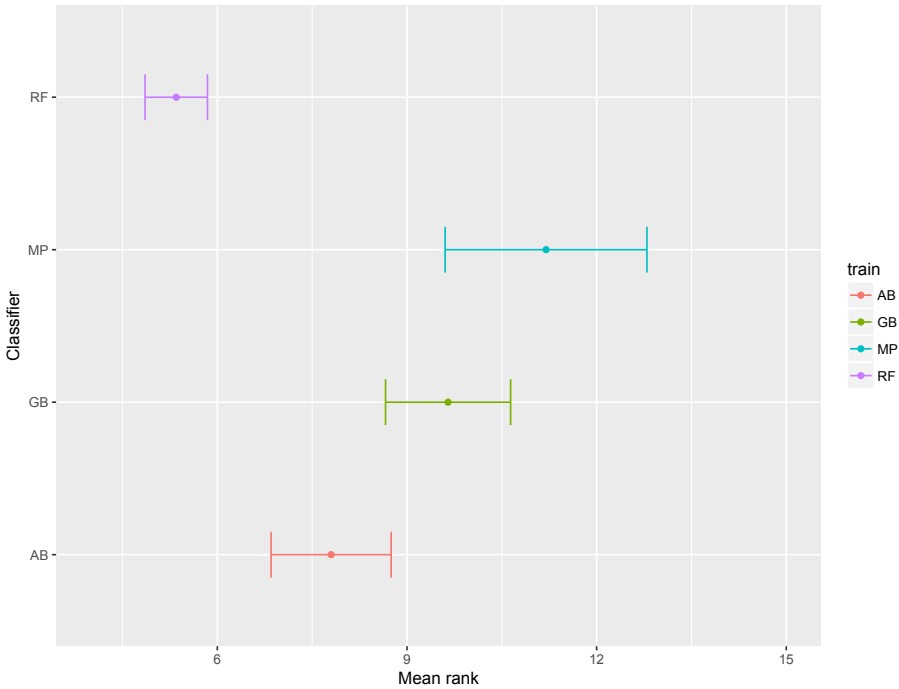

**Figure 5  Mean rank per classifier.** The dot represents the mean rank and bars represent standard error. Lower ranks indicate better performance in terms of AUPRC. Classifiers are indicated by AB, adaptive boosting; GB, gradient boosting; MP, multilayer perceptron; RF, random forest.

table produced by sRNACharP and calculates the probability of being a bona fide sRNA for each sRNA included in the feature table. sRNARanking is available at https://github.com/BioinformaticsLabAtMUN/sRNARanking.

## Attribute importance

Based on the mean decrease in accuracy estimated by the random forest algorithm, all attributes contribute positively to obtain a more accurate model (Fig. 7). The seven attributes clustered in three levels of importance: those with a mean decrease in accuracy greater than 20; those with a mean decrease in accuracy between 10 and 15, and those with a mean decrease in accuracy lower than 10. The most important attributes are the distance to the closest ORFs and the distance to the closest predicted Rho-independent terminator. The two attributes that seem to contribute the least to the accuracy of a model are the Boolean features indicating whether or not a genomic sequence is transcribed on the same strand as its closest ORFs.

## Assessing the performance of our best model on benchmark datasets

*Lu, Goodrich-Blair & Tjaden (2011)* evaluated the performance of four comparative genomics-based leading tools for sRNA prediction on a dataset composed of sRNAs from 14 different bacteria (Table 2), and found recall rates of 20%–49% with precisions of 6%–12%; while our RF-combined model (sRNARanking) achieved precision rates of 85%
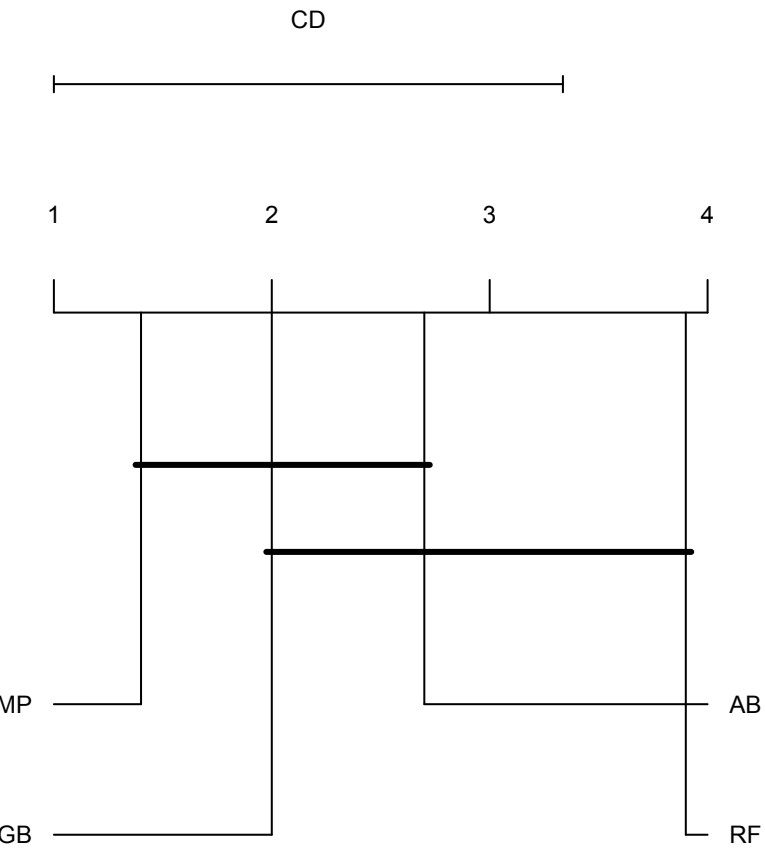

**Figure 6 Critical difference plot.** Classifiers that are not deemed significantly different by the Nemenyi test at a significance level of 0.05 are connected. The methods with the lowest (best) ranks are to the right.

to 96% at those same recall rates (Fig. 8). Our approach also outperformed Barman et al.'s SVM method on this dataset (Fig. 8).

*Arnedo et al. (2014)* and *Barman, Mukhopadhyay & Das (2017)* evaluated the performance of their methods in terms of sensitivity (recall) and specificity. Figure 9A shows the Sensitivity-Specificity curve of sRNARanking and Barman et al.'s SVM method on the SLT2 dataset. When both methods are compared with random genomic sequences as negative instances our best model sRNARanking outperformed Barman et al.'s SVM method in terms of sensitivity, specificity and AUPRC (Fig. 9B). Barman et al.'s SVM method obtained better performance with shuffled genomic sequences as negative instances.

The main goal of this study was to precisely rank sRNAs detected from RNA-seq data to guide further experiments to functionally characterize sRNAs. If we assume that microbiologists would only select a few of the top-scoring predictions for Northern blot validation, then at a sensitivity (recall) level of 10%, sRNARanking has a precision of 90% on the SLT2 dataset (Fig. 9B). In other words, if we assume that the top 10% predictions would be selected for Northern blot validation, only two out of 18 candidate sRNAs would fail to be detected.

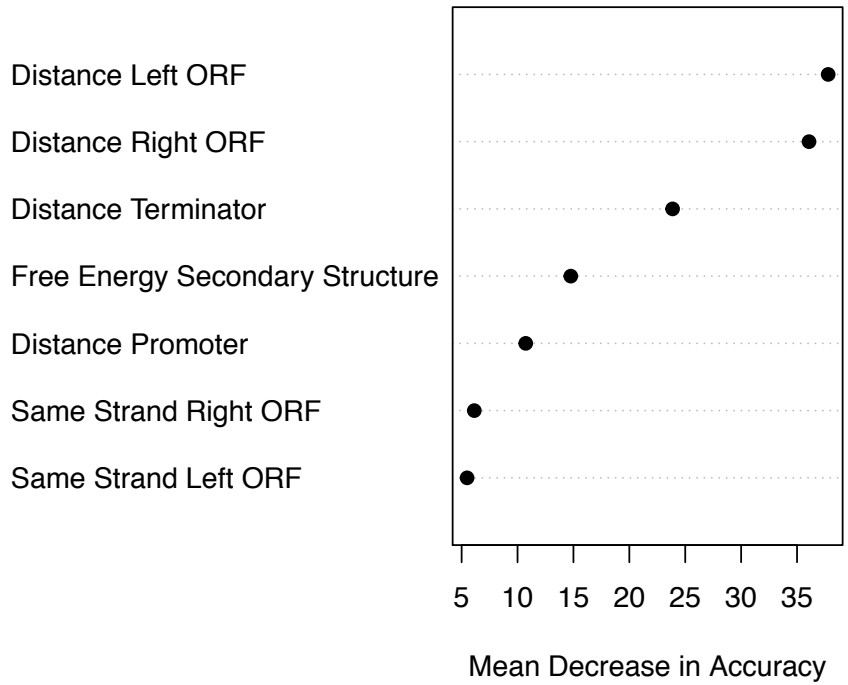

**Figure 7 Attribute importance.** Mean decrease in accuracy per attribute as estimated by the random forest algorithm. Attribute importance is plotted on the *x*-axis. Attributes are ordered top-to-bottom as most- to least-important. Three levels of importance are observed: high importance attributes (distances to closest ORFs and distance to terminator); medium importance attributes (free energy of secondary structure and distance to promoter), and low importance attributes (same strandness as closest ORFs).

### Varying ratio of positive to negative instances

We expected that constructing the models with a slightly more unbalanced training data set (with a ratio of 1:3 positive to negative instances) than those datasets previously used would have an impact on the prediction performance of the models. To explore whether or not this hypothesis was correct, we constructed two additional RF-Combined models one with a ratio of 1:1 positive to negative training instances and another one with a ratio of 1:2 positive to negative training instances. The positive instances of all three models were the same, and the negative instances of the more balanced datasets were proper subsets of the less balanced data sets. That is, all negative instances in the 1:1 training set were contained in the 1:2 training set and in the 1:3 training set. Contrary to our hypothesis, our results suggest that varying the ratio of positive to negative instances from 1:1 to 1:3 does not have a major impact in the prediction performance of the RF-Combined model (Figs. S2–S8).

### DISCUSSION

We anticipated that the distance to the closest promoter, the distance to the closest terminator and the energy of the secondary structure would be the most important attributes to predict whether or not a genomic sequence is a bona fide sRNA. As promoters determine when and how transcription of a nearby gene is initiated (*Alberts et al., 2002*),

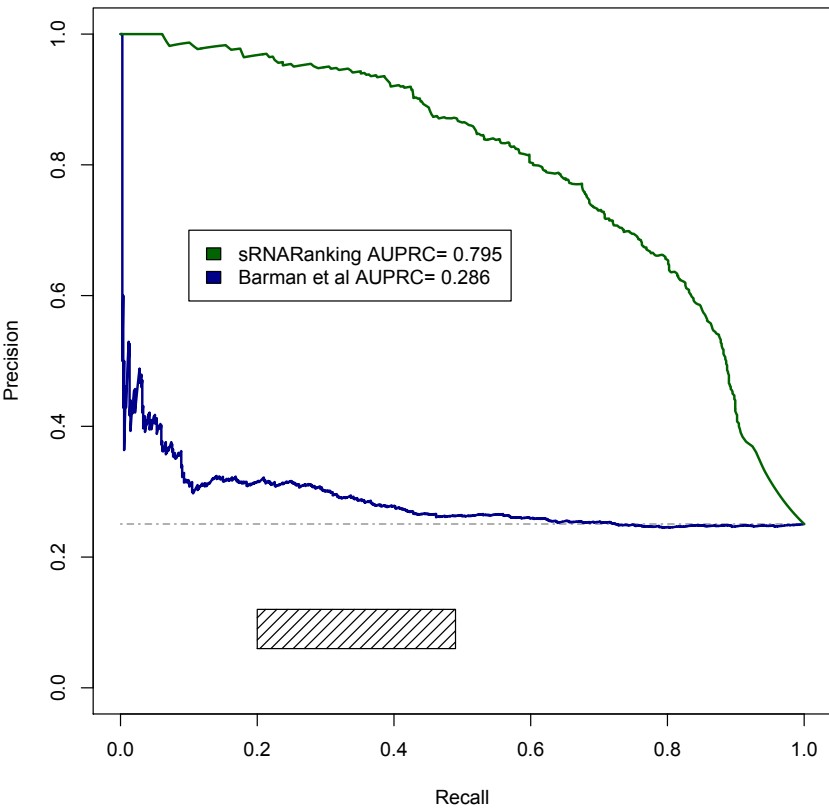

**Figure 8  PRC of sRNARanking and Barman et al.'s SVM method performance on Lu et al.'s multi-species dataset.** The four approaches assessed by Lu et al. achieved recall of 0.20 to 0.49 with precision of 0.06 to 0.12. The corresponding area in the PRC is indicated by the rectangle. The horizontal dashed line indicates the performance of a random classifier.

we expected genuine sRNAs to be close to a promoter. However, as current best programs to predict promoters have a recall rate between 49% and 59% (*Shahmuradov et al., 2017*), many sRNAs might incorrectly be represented as not having a promoter nearby when in fact they do. Gene expression in bacteria is also regulated by termination of transcription, often in response to specific signals (*Santangelo & Artsimovitch, 2011*). Being in proximity to a Rho-independent terminator is used as evidence for genome annotation (*Nikolaichik & Damienikan, 2016*). However, bacteria commonly regulate gene expression by using *cis*-acting RNA elements for conditional transcription termination (*Dar et al., 2016*). These *cis*-acting terminators are not predicted by TransTermHP (*Kingsford, Ayanbule & Salzberg, 2007*), and thus many sRNAs might incorrectly be represented as not being in the proximity to a terminator. We expect that improving bacterial promoter and terminator prediction will increase the importance of these features and improve sRNA prediction using genomic context features by reducing the number of false negatives made by sRNARanking. Many sRNAs have a stable secondary structure; however, sRNAs are also known to show heterogenous structures (*Wagner & Romby, 2015*). This might reduce the importance of

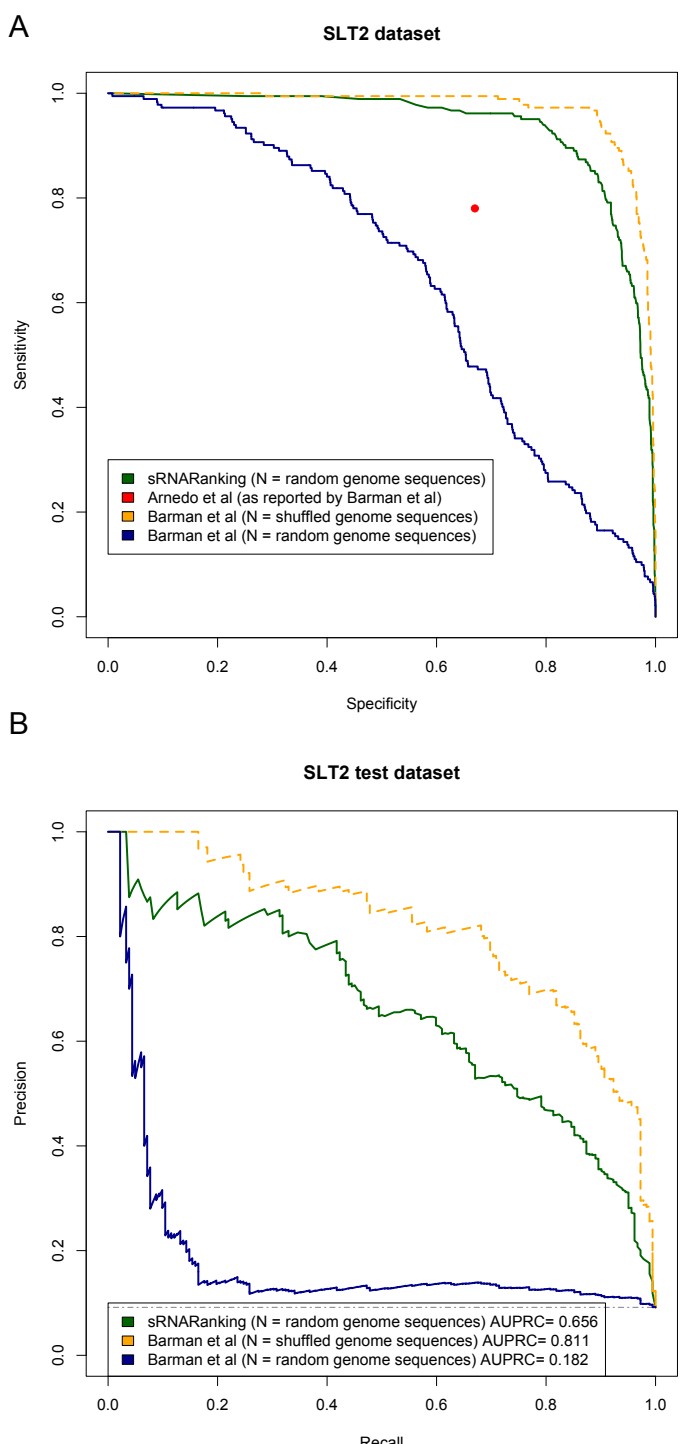

**Figure 9 Methods performance on the SLT2 dataset.** Barman et al.'s (*2017*) SVM method performance is reported on Barman et al.'s test set with shuffled sequences (artificial sequences) as negative instances and on our test set with random genomic sequences (real genomic sequences) as negative instances. (A) Sensitivity-Specificity curve. Arnedo et al.'s approach reported sensitivity and specificity is indicated with a red dot. (B) Precision-Recall curve. The horizontal dashed line indicates the performance of a random classifier.

the energy of the secondary structure as a feature to predict sRNAs. We believe that the distances to the closest ORFs are the most important attributes partially due to a bias in the training data. 93% of the negative instances (random genomic sequences) in the combined training data overlap the two neighbouring ORFs (i.e., their distances to their closest ORFs are zero), while 70% of the positive instances (bona fide sRNAs) are intergenic (i.e., their absolute distances to their closest ORFs are greater than zero). This bias in the data may be corrected as more antisense sRNAs (asRNAs) and partially overlapping sRNAs are experimentally verified as bona fide sRNAs.

We hypothesized that *R. capsulatus* training data produced worse performing models because it includes as positive instances a higher number of non-intergenic sRNAs (18 or 50%). In fact, the best performing model obtained lower AUPRC for *R. capsulatus* and *E. coli* test datasets (Figs. S2–S6). These two bacterial species have the higher proportion of non-intergenic bona fide sRNAs: 51% and 40% of the bona fide sRNAs of *R. capsulatus* and *E. coli*, respectively, overlap neighbouring ORFs; while 17.4%, 26.5% and 36.8% of the bona fide sRNAs of *S. pyogenes*, *S. enterica* and *M. tuberculosis*, respectively, overlap neighbouring ORFs. Additionally, 17 *R. capsulatus* putative sRNAs included as positive instances were found to be conserved in the genome of at least two other bacterial species but have not been verified in the wet lab. Some of these 17 putative *R. capsulatus* sRNAs chosen as positive instances based on sequence conservation may actually be false positives. *Barman, Mukhopadhyay & Das (2017)* also observed that the performance of their approach for predicting *E. coli* sRNAs was inferior than the performance obtained for other bacteria. They suggested that a reason for this might be the higher number of experimentally verified sRNAs of *E. coli* overlapping with ORFs (*Barman, Mukhopadhyay & Das, 2017*).

With respect to the different machine learning approaches assessed, RF seems to be better suited for the task of prioritizing bona fide sRNAs than the other four classifiers (AB, GB, MP and LR). Statistical tests results supported this by deeming the difference in performance between the models obtained by RF and models obtained by the other classifiers as statistically significant. To be able to use deep learning for sRNA prioritization, datasets at least one order of magnitude larger than the ones currently available are required.

To demonstrate the ability of the models to generalize to other bacterial species, we validated the models on data from bacterial species that were not part of the training set. In fact, using data from the same bacterial species on the training and test sets was not a factor to explain variance in model performance. This indicates that models are able to learn sRNAs features that are species independent, and even taxa independent as the AUPRC values obtained in the *M. tuberculosis* and Lu et al.'s (*2011*) test sets suggest. Using data from different bacterial species and experimental conditions is expected to lead to improved predictive models. In fact, training the classifiers with the combined data generated models that either outperform, or were comparable to, the models obtained from training the classifiers with data from a single bacterium. To allow other researchers to rank their own sRNAs, we have implemented sRNARanking, an R script containing the RF-Combined model.

To compare our best performing model with current approaches, we evaluated sRNARanking on a multi-species dataset (*Lu, Goodrich-Blair & Tjaden, 2011*) and

demonstrated that sRNARanking clearly outperformed an SVM-based approach using tri-nucleotide composition features and four comparative genomics-based approaches (Fig. 8). Additionally, we compared sRNARanking performance on a SLT2 dataset with two recently published approaches: a meta-approach (*Arnedo et al., 2014*) and a SVM-based approach (*Barman, Mukhopadhyay & Das, 2017*). sRNARanking achieved better performance than both approaches (Fig. 9). Our results also suggest that using real genomic sequences as negative instances gives a more conservative predictive performance estimate than using artificial (shuffled) genomic sequences as negative instances. Although sRNARanking outperformed the other published methods in the benchmark datasets, there is still room for improvement of computational identification of sRNAs from genomic sequences.

A multitude of sRNAs have been detected in many bacterial species. The sheer number of novel putative sRNAs reported in the literature makes it infeasible to validate each of them in the wet lab. Thus, there is the need for computational approaches to characterize putative sRNAs and to rank these sRNAs on the basis of their likelihood of being bona fide sRNAs. Our results demonstrate that a RF-based approach using genomic context and structure-based features is able to detect intrinsic features of sRNAs common to a number of bacterial species, overcoming the challenge of the low sequence conservation of sRNAs. As the number of detected sRNAs continues to raise, computational predictive models as the one here presented will become increasingly valuable to guide further investigations.

## Abbreviations

| | |
|---|---|
| **LR** | logistic regression |
| **MP** | multilayer perceptron |
| **AB** | adaptive boosting |
| **GB** | gradient boosting |
| **RF** | random forest |
| **FDR** | false discovery rate |
| **AUC** | area under receiver operating characteristic curve |
| **AUPRC** | area under the precision–recall curve |
| **LOO CV** | leave-one-out cross-validation |
| **ORF** | open reading frame |
| **nts** | nucleotides |
| **sRNA** | small non-coding RNA |

## ACKNOWLEDGEMENTS

We thank Emilio Palumbo for providing technical support for the implementation in Nextflow, Dr. Meruvia-Pastor for providing feedback on the manuscript, and the peer reviewers whose comments have greatly improved this manuscript.

### Funding

This work was supported by a Discovery Grant (No. 402087-2011) of the Natural Sciences and Engineering Research Council of Canada (NSERC) to Lourdes Peña-Castillo. The funders had no role in study design, data collection and analysis, decision to publish, or preparation of the manuscript.

### Grant Disclosures

The following grant information was disclosed by the authors:
Discovery Grant: 402087-2011.
Natural Sciences and Engineering Research Council of Canada (NSERC).

### Competing Interests

The authors declare there are no competing interests. Lourdes Peña-Castillo is an Academic Editor for PeerJ.

### Author Contributions

- Erik J.J. Eppenhof performed the experiments, contributed reagents/materials/analysis tools, prepared figures and/or tables, authored or reviewed drafts of the paper, approved the final draft.
- Lourdes Peña-Castillo conceived and designed the experiments, performed the experiments, analyzed the data, contributed reagents/materials/analysis tools, prepared figures and/or tables, authored or reviewed drafts of the paper, approved the final draft.

### Data Availability

Code is available at GitHub:
https://github.com/BioinformaticsLabAtMUN/sRNACharP
https://github.com/BioinformaticsLabAtMUN/sRNARanking
Data and code are available as a Supplemental File.

### Supplemental Information

Supplemental information for this article can be found online at http://dx.doi.org/10.7717/peerj.6304#supplemental-information.

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
