# Peer review of "Prioritizing bona fide bacterial small RNAs with machine learning classifiers"

_PeerJ, doi:10.7717/peerj.6304_

## Round 0.1 · original submission · Major Revisions

I have considered your appeal. Thank you for clarifying your method of selecting negative examples. Based on the explanation and materials that you have provided, I will change the decision to Major Revisions.

Note that Reviewer #3's review does include a correct interpretation of your negative selection process and that the reviewer still has concerns about the adequacy of the selected negatives. Please carefully consider the concerns of the reviewers as you make the revisions.

To consider this for publication in PeerJ, I need to be convinced that the selection of negatives provides a high enough standard for the findings to be robust and scientifically sound. This is likely to include additional analyses that support the validity of this work.

· Appeal

Appeal

Thank you for reviewing our manuscript and notifying us about the editorial decision. However, the concern that leads Dr. Greene to reject our manuscript, namely that "the negative samples generated via random shuffling of the input sequence may not provide an adequate control for small RNA discovery", is unfounded. We did not use as negative instances shuffled versions of the input sequences, instead we used genomic sequences randomly generated using BedTools (RE: lines 72-74 of our manuscript). We used exactly what the reviewers wanted us to use as negative instances. I now recognize that the BedTools utility we used (shuffleBed) has a name that may lead a reviewer to misunderstand what the tool does. However, this tool shuffleBed (http://bedtools.readthedocs.io/en/latest/content/tools/shuffle.html) generates a random instance for each genomic coordinate given on a random chromosome at a random position. The size and strand of each feature are preserved.

In view of this clarification that resolves the main concern of Dr. Greene, would you reconsider the decision?


· · Academic Editor

Reject

The reviewers noted some strengths of your manuscript and identified small RNAs as an interesting topics. However, there were also weaknesses. The reviewers expressed concerns about the selection of the training and evaluation sets of small RNAs. The concern that leads me to reject this manuscript is that the negative samples generated via random shuffling of the input sequence may not provide an adequate control for small RNA discovery.

# ·

Basic reporting

The manuscript describes an exploration of machine learning, specifically classification, methods to predict novel small non-coding RNAs (sRNAs). Using seven features defined for sRNAs and several well-established classification algorithms, the authors rigorously evaluated how well these algorithms perform for sRNA prediction. An encouraging aspect of this evaluation is that training and validation examples, albeit small in number, are drawn from multiple bacterial species, including a validation set from a species not in the same taxa. The results show that Random Forest (RF) is the best performing algorithm overall, with different features contributing to various levels in this performance.

The most general weakness of the manuscript is the lack of biological justification/explanation of several aspects of the study on an otherwise biologically important problem, such as the following:
• A justification of why these seven features were selected and not others.
• An explanation of the observed importance of the features in the RF model (Fig 6) beyond the data availability arguments presented in the Discussion.
• The significance of validation sets from the same and different taxa isn’t discussed until page 11 (lines 322-3). Even here, it’s unclear why the results for Ec are worse than those for Mt given the above differences in taxa.
These and other biological aspects of the study need to be better explained/justified.

The manuscript is generally well-written, except for minor errors like sevens (should be “seven”) and exploited (use a more positive word) in the abstract itself. Also, the term “data” should be treated as plural everywhere, not singular (e.g. data are/were, not is/was). The manuscript will do well with another round of close proof-reading.

Experimental design

The weakest aspect of the study is the small size of the training and validation sets (Table 2) the conclusions are based on. While it is understandable that it is difficult to get reliable data for a relatively novel problem like this, appropriate precautions like the following should be followed when reporting results:
• Given that the training and validation sets are already small and randomly determined to begin with (Table 2), it is imperative to create multiple such training and validation sets and all reported results must be shown as averages along with error bars, e.g., in Figure 2. From the first paragraph of the Performance Assessment section, it appears that an effort is made in this direction, but not clearly stated, and is not adequately reflected in the results, which is a must.
• As noted by the authors themselves, AUC is not an appropriate evaluation measure for the unbalanced training and validation sets employed in the study. Still, AUC is used as the main evaluation measure (Fig 2), as well as the measure optimized in various classification models. The same processes should be conducted using the precision-recall-Fmeasure measures, which are much more appropriate for such situations, and the consistency of the results should be rigorously examined. Refer to the following articles for details:

Lever, J., Krzywinski, M. & Altman, N. Points of Significance: Classification Evaluation. Nature methods 13, 603–604 (2016).

Saito, T. & Rehmsmeier, M. The precision-recall plot is more informative than the ROC plot when evaluating binary classifiers on imbalanced datasets. PLoS One 10, e0118432, https://doi.org/10.1371/journal.pone.0118432 (2015).

• The training sets are arbitrarily designed to have 3 times the number of negative examples as compared to the positive ones. Sensitivity analyses should be conducted to assess the dependence of the results on this ratio.
• The authors have taken the appropriate step of testing the statistical significance of classifier performance differences. However, the Friedman-Nemenyi tests are recommended to be more appropriate for this testing in Section 3.2 of this paper: Demsar, J. Statistical Comparisons of Classifiers over Multiple Data Sets. J. Mach. Learn. Res. 7, 1–30 (2006). The Critical Difference (CD) plots discussed in this paper are also a better way to visualize the results of such tests than Table 2 of the manuscript. It’ll be useful to test the validity of the current results based on the Friedman-Nemenyi tests and visualize the results as CD plots.
• Finally, the purpose of violin plots in Figure 3 is unclear and not adequately explained. Better visualization(s) and/or explanation(s) should be used in the next version.

Validity of the findings

The findings are generally valid, but must be strengthened based on the points raised above.

Additional comments

Addressing the above comments should strengthen the manuscript and make it more acceptable for publication.

Reviewer 2 ·

Basic reporting

Overall, the manuscript was well written. I did notice a few typographical errors.

The Introduction was relatively short. It would be helpful if the authors provided additional context on identifying bacterial sRNAs, in general. What research applications exist after these have been identified? Have any other researchers developed bioinformatic algorithms for prioritizing these? If so, what techniques did they use and how do those differ from your own? Also, why were these specific sequence properties selected for the machine-learning analysis?

I appreciate that the authors provided source code and have made it easy for others to execute their software! I am not sure I understand why the provided software for ranking sRNAs is in the R language whereas their analysis was performed in Python. The outputs should be similar, but it would help to at least provide an explanation about this, if not redo the analysis in the paper using R.

Along those lines, the authors should provide the specific code (and raw data) that they used in their analyses.

Figure 2 conveys the message that AUC values differed depending on the species and that LR performs relatively poorly. But the large number of bars makes it difficult to discern these conclusions (and perhaps others that could be discerned). Perhaps using violin plots like Figure 3 would be better than bar plots.

Experimental design

I am unsure about the way that the negative instances were generated. In my understanding, the authors shuffled sequences from the positive instances. Are these sequences different enough from what you would see in a "real" genome that it would be easy for an algorithm to distinguish between the positive and negative sequences? Wouldn't these negative sequences have the same properties as the positive sequences, at least for some of these properties (distance to the closest predicted promoter site, distances to ORFs, etc.)? Or was genomic position somehow randomized so that these would differ? Not enough details are provided to judge whether this was performed in a reasonable way.

The dataset has a 3:1 ratio of positive to negative instances. Why was this ratio selected? Might a more (or less) extreme ratio better represent reality?

The authors state that they performed "five training runs" for each combination of classification algorithm and data set. Previously, they state that the training sets had 80% of the total data. In what way did each of these training runs differ from each other? Did they use different subsets of the original training sets? Or were they different 80/20 splits of the full data set? If it is the latter, then this could significantly bias the results. In addition, the manuscript states that "The five training runs were done after optimizing the classifiers' parameters with LOOCV." Please be more clear in describing what data were used for this optimization.

Why were no E. coli or M. tuberculosis data used in the training sets? I'm not saying it should, but I didn't see any justification for this stated in the manuscript.

On line 87, it says that a range of [-150, length of the sequence] was used. But if there was no promoter in that region, then a value of -1000 was used. Perhaps I am misunderstanding, but why not just use a value of -1000 from the start, as is indicated on line 89?

I am unsure why the authors used sklearn for all the algorithms except Multilayer Perceptron. sklearn has an implementation of Multilayer Perceptron.

Did the authors generate probabilistic predictions? Or binary predictions?

Validity of the findings

I am unclear on why the authors used the metrics that they did to evaluate algorithm performance. They state that the AUC is susceptible to imbalanced data. Although that may be true in some extreme cases, it is much better than most other metrics for imbalanced data. The authors also use a few combinations of precision and recall. They evaluated prediction using recall rates of 10%, 40%, and 60%. It seems that 90% should also be used to make this symmetrical, but the authors did not state anything about this. Furthermore, why not use AUPRC instead of of using these arbitrary thresholds?

The Discussion and Conclusion were fine, but I'd have to have answers to some of my previous questions to fully evaluate the validity of these statements.

Additional comments

The authors have used machine-learning to predict whether small bacterial RNAs are putative based on properties of DNA sequences. This is a clever approach because they rely not on the specific sequence but rather on metadata about the sequences, thus potentially providing better generalizability.

Reviewer 3 ·

Basic reporting

References: The Random Forests method was proposed by Breiman. The other references provided for the method are not the random forests method. In the context of the low dimensional data used by the authors it probably doesn't matter though, since they use all features in every tree.

Experimental design

No comment

Validity of the findings

1. Identification of sRNAs is an active area of research, and many methods for this task have been proposed. There is no comparison of the authors' results with any other published method, so it is not possible to assess their contribution.

2. The authors use a few easy to compute features. Existing methods use a wealth of other information, including comparison to related bacteria, base composition information. Several methods predict sRNAs and their targets at the same time to obtain increased accuracy. See e.g.
Pichon, Christophe, and Brice Felden. "Small RNA gene identification and mRNA target predictions in bacteria." Bioinformatics 24.24 (2008): 2807-2813.

3. The accuracy of the method is likely not sufficient to be useful for applying the method on a genome-wide scale, where even a small false positive rate can lead to many erroneous predictions.

4. It's not clear how the negative examples are generated - the authors state that their sequences are shuffled versions of the original sequences, but it's not clear how the location is chosen, since that has an effect on the other features. Simply shuffling is potentially producing "easy" negative examples, since those are sequences that do not necessarily have the characteristics of real genomic sequences. Random sequences from the genome are likely a better choice; some of them might be sRNAs, but the contamination level is likely quite small, and would not present an issue.

5. Generating p-values for comparing classifiers is a tricky proposition. See:
Salzberg, Steven L. "On comparing classifiers: Pitfalls to avoid and a recommended approach." Data mining and knowledge discovery 1.3 (1997): 317-328.

Additional comments

Another minor comment:
Scientific notation: 2e-16 is 2*10^(-16) and not 2*e^(-16)!

---

## Round 0.2 · Major Revisions

Reviewers #1 and #3 have additional points that should be addressed. In particular, please provide the literature support for the assertions noted by reviewer #1 and please address the issues raised with the validity concerns of reviewers #1 and #3.

Please also carefully define what you mean "validation" sets as noted by reviewer #3. Fields vary somewhat in terms of how this language is used (i.e., in some validation conveys an independent validation set, while in ML validation is not as strong as test).

·

Basic reporting

The authors have made a substantial effort to address the reviewers’ comments, but several issues still remain:

1. The reasoning for the selection of the features on lines 68-71 should be supported by data and/or references.
2. The authors state that the recall rate of promoter prediction programs isn’t very high. So, what is the impact of this on the sRNA prediction results? Similar issues exist presumably for the other feature extractors as well. What is the impact of those?
3. All the statements added in lines 370-390 to explain the observed importance of the features in the RF model must be supported by data and/or references.
4. The fonts in Figure 4 are too small to be readable.
5. The term “FDR corrected pvalue” is redundant – FDR is corrected pvalue, but it should also be mentioned how the FDR is obtained.
6. The forward slashes in Figure 8 is an atypical way of showing the variation. Why not show a curve, or a region, whatever is appropriate?

Experimental design

1. The reasoning for only creating a 1:3 unbalanced dataset isn’t satisfactory, especially because this factor is expected to have a major impact on the prediction performance. So, this ratio has to be varied to measure this impact.

Validity of the findings

1. As stated on line 361, “sRNARanking’s specificity of 88% is comparable to the 91% specificity of Barman et al’s approach at a sensitivity of 85%.”. So, what is the benefit of this study compared to Barman’s?
2. Why isn't LR included in the CD plot in Figure 6? Also, unless there is a novel interpretation of the CD plot (the order seems to be reversed), RF seems like the worst performer instead of the best, i.e., is the rightmost entry instead of the expected leftmost.
3. The utility of the violin plot in Figure 3 is still unclear. Typically, when such a plot is shown, there is a variable on the Y-axis, e.g. individual genes’ expression, and the plots show the dependence of this value on the factors listed on the X-axis. So what is the parallel interpretation here? Are the authors calculating a per-example AUPRC? This has to be explained clearly when Figure 3 is mentioned on line 309.

Reviewer 2 ·

Basic reporting

I have no additional comments.

Experimental design

I have no additional comments.

Validity of the findings

I have no additional comments.

Reviewer 3 ·

Basic reporting

The authors have addressed most of the concerns raised in my initial review. The comparison with existing methods improved the manuscript significantly. However, several comments remain.

Experimental design

- The authors added a comparison with other published methods. Although the same positive examples were used, it's not clear if the method for choosing negative examples, and their numbers were the same.

- I agree with reviewer Pandey remark about the evaluation procedure employed by the authors, and have additional comments in that regard. In machine learning the term "validation set" refers to data that is used for choosing classifier hyperparameters. The authors appear to use this term to describe the test set on which performance is evaluated. Furthermore, the fact that the authors mention specific hyperparameter values as optimal when discussing the various classifiers is inconsistent with the mention that multiple training sets were used, as each choice of training set might end up with different optimal hyperparameters.
The model selection/evaluation proposed by the reviewer Pandey makes a lot more sense than the one used by the authors in view of the small number of examples. Measuring the variability of the performance over several training runs has limited usefulness, and in algorithms such as logistic regression which has a single global optimal solution that is even more the case. Finally, using LOO cross validation for parameter selection is clearly overkill. Nested cross-validation is the standard technique for classifier evaluation and model selection in the area of machine learning, and is well supported by packages such as scikit-learn. The following paper is a good reference on this topic, and goes even further, suggesting repeated nested cross validation as a good approach for small datasets:
https://jcheminf.biomedcentral.com/articles/10.1186/1758-2946-6-10

Validity of the findings

- In my initial review I mentioned the fact that the method is likely of limited use when applied genome-wide, because it is likely to produce a large number of false positives.
My comment was:

The accuracy of the method is likely not sufficient to be useful for applying the method on a genome-wide scale, where even a small false positive rate can lead to many erroneous predictions.

The answer of the authors was:

As we demonstrated in the revised version of the manuscript, our method outperforms or is comparable to existing approaches in terms of precision/recall and sensitivity/specificity.

While that may be true, it does not refute my initial comment. It is very unfortunate that researchers developing methods for sRNA detection choose to use relatively balanced datasets while testing their methods. This evaluation methodology gives no indication of method usability when used genome-wide (area under the precision recall curve is not invariant to the ratio of positive to negative examples), and can give biologists the wrong impression about the usability of the method. Furthermore, the majority of the features computed by the authors' method (except for the free energy feature) are insensitive to shifting the region over which they are computed. That is another reason for the limited usability genome-wide. Perhaps when combined with a method such as Barman et al, which uses primary sequence, more specificity can be obtained. Overall, it appears like the problem of sRNA identification directly from genomic sequence is a hard unsolved problem, and this should be indicated.

Additional comments

Minor comments:

- In line 100 n is used to denote the number of negative examples, but in lines 102-105, n is used differently to denote the relative fraction of negative to positive examples.

- In the description of logistic regression you reference "balanced" mode of some implementation of LR, but no mention is made which software was used. For the other classifiers specific software/version should be mentioned as well.

---

## Round 0.3 · accepted · Accept

I have read the previous reviewer concerns again, as well as the response and I find that the reviewer concerns have been adequately addressed by the revision. I noticed, on my read through the revised manuscript, that some of the revisions may need to be carefully proofread. For example, "shuffling the sequences bases" should probably have sequences as possessive ("sequences' bases'). Please do one more proofread during the production process.

#

---

## Author Rebuttal · Round 0.3

**Dear Dr. Greene**
**We appreciate the reviewers' efforts and comments on the submitted manuscript. Our detailed response to their comments is below.**

**Editor comments (Casey Greene)**
Reviewers #1 and #3 have additional points that should be addressed. In particular, please provide the literature support for the assertions noted by reviewer #1 and please address the issues raised with the validity concerns of reviewers #1 and #3.

> **We have provided the literature support and addressed the issues raised with respect to the validity of the study.**

Please also carefully define what you mean "validation" sets as noted by reviewer #3. Fields vary somewhat in terms of how this language is used (i.e., in some validation conveys an independent validation set, while in ML validation is not as strong as test).

> **We are now employing the term "test set" instead of "validation set", and have given a definition on line 243 of the meaning of "test set".**

**Reviewer 1 (Gaurav Pandey)**
Basic reporting
The authors have made a substantial effort to address the reviewers' comments, but several issues still remain:

1. The reasoning for the selection of the features on lines 68-71 should be supported by data and/or references.

> **References have been added.**

2. The authors state that the recall rate of promoter prediction programs isn't very high. So, what is the impact of this on the sRNA prediction results? Similar issues exist presumably for the other feature extractors as well. What is the impact of those?

> **We believe the impact is an increase of the number of false negatives of our model. Once better feature extractors become available one will be able to quantify their impact.**

3. All the statements added in lines 370-390 to explain the observed importance of the features in the RF model must be supported by data and/or references.

> **References have been added.**

4. The fonts in Figure 4 are too small to be readable.

> **The font size has been increased.**

5. The term "FDR corrected pvalue" is redundant – FDR is corrected pvalue, but it should also be mentioned how the FDR is obtained.

> **Reference has been added in lines 276-278. For clarity sake, we prefer to use the term FDR-corrected pvalue.**

6. The forward slashes in Figure 8 is an atypical way of showing the variation. Why not show a curve, or a region, whatever is appropriate?

**Lu et al (2011) provided a performance range of four methods. Thus, we are not showing variation, we are showing the range as reported by Lu et al. Nevertheless, we have replaced the forward slashes with a distinctive rectangle.**

## Experimental design

1. The reasoning for only creating a 1:3 unbalanced dataset isn't satisfactory, especially because this factor is expected to have a major impact on the prediction performance. So, this ratio has to be varied to measure this impact.

**We have varied the ratio and found that it does not have a major impact on the prediction performance of sRNARanking. A new subsection in Results presenting this finding, and additional figures 2-8 have been included, that show the PRC of RF-Combined models constructed with different ratio of positive to negative examples on all test datasets.**

## Validity of the findings

1. As stated on line 361, "sRNARanking's specificity of 88% is comparable to the 91% specificity of Barman et al's approach at a sensitivity of 85%.". So, what is the benefit of this study compared to Barman's?

**We have now executed Barman et al's approach with exactly the same test datasets (Lu's and SLT2) and our method clearly outperforms Barman et al's approach. We have updated our manuscript to reflect this (lines 290-300 and 365-375). The benefit is that we are showing that better predictive performance can be obtained using a set of genomic context features. Additionally, as mentioned by reviewer 3, this study might motivate the development of another method combining ours and Barman's.**

2. Why isn't LR included in the CD plot in Figure 6? Also, unless there is a novel interpretation of the CD plot (the order seems to be reversed), RF seems like the worst performer instead of the best, i.e., is the rightmost entry instead of the expected leftmost.

**We are not using a novel interpretation of the CD plot. In fact, we are using the same interpretation as in the original manuscript that introduced CD plots ( *Statistical Comparisons of Classifiers over Multiple Data Sets,* Demsar, 2006). As it is explained by Demsar (page 15) "The axis is turned so that the lowest (best) ranks are to the right since we perceive the methods on the right side as better".**
**As LR was clearly outperformed by the other four classifiers, we excluded LR results from further analysis (already explained in lines 322-325).**

3. The utility of the violin plot in Figure 3 is still unclear. Typically, when such a plot is shown, there is a variable on the Y-axis, e.g. individual genes' expression, and the plots show the dependence of this value on the factors listed on the X-axis. So what is the parallel interpretation here? Are the authors calculating a per-example AUPRC? This has to be explained clearly when Figure 3 is mentioned on line 309.

**The variable on the Y-axis is the AUPRC per model on a specific test set. For each classifier, four models were generated (one for each training set). Each of these models was evaluated**

**on each of the five test sets. We are showing the dependence of the AUPRC of the models on the classifier used to construct the models. This has been explained in lines 320-322.**

**Reviewer 2 (Anonymous)**
**Basic reporting**
I have no additional comments.
**Experimental design**
I have no additional comments.
**Validity of the findings**
I have no additional comments.

**Reviewer 3 (Anonymous)**
**Basic reporting**
The authors have addressed most of the concerns raised in my initial review. The comparison with existing methods improved the manuscript significantly. However, several comments remain.
**Experimental design**
- The authors added a comparison with other published methods. Although the same positive examples were used, it's not clear if the method for choosing negative examples, and their numbers were the same.

> **We have now executed Barman et al's approach with exactly the same test datasets (Lu's and SLT2)  and the new results are included in the manuscript.**
> **As already explained in the previous version of the manuscript (lines 109-114), we are not using the same method for choosing negative examples. Arnedo et al and Barman et al used shuffled genomic sequences while we are using sequences from random genomic locations. In other words, our negative instances are real genomic sequences present in the bacterial genomes, while Barman et al's negative instances are artificial sequences not present in the bacterial genomes.**

- I agree with reviewer Pandey remark about the evaluation procedure employed by the authors, and have additional comments in that regard. In machine learning the term "validation set" refers to data that is used for choosing classifier hyperparameters. The authors appear to use this term to describe the test set on which performance is evaluated.

> **We are now employing the term "test set" instead of "validation set", and have given a definition on line 243 of the meaning of "test set".**

Furthermore, the fact that the authors mention specific hyperparameter values as optimal when discussing the various classifiers is inconsistent with the mention that multiple training sets were used, as each choice of training set might end up with different optimal hyperparameters.

> **In the manuscript in lines 167-172, we have explained that "For each classifier, the 'best' parameters were obtained by maximizing the average area under the ROC curve (AUC) when performing leave-one-out cross-validation (LOO CV) on the training data (Fig.2). The final 'best' parameters per classifier used were those given the largest average AUC."**

The model selection/evaluation proposed by the reviewer Pandey makes a lot more sense than the one used by the authors in view of the small number of examples. Measuring the variability of the performance over several training runs has limited usefulness, and in algorithms such as logistic regression which has a single global optimal solution that is even more the case.

**As our results show, classifiers have large variability of performance over several training runs; thus we still believe that having an understanding of the variability of the classifiers over several training runs is useful.**

Finally, using LOO cross validation for parameter selection is clearly overkill. Nested cross-validation is the standard technique for classifier evaluation and model selection in the area of machine learning, and is well supported by packages such as scikit-learn. The following paper is a good reference on this topic, and goes even further, suggesting repeated nested cross validation as a good approach for small datasets: https://jcheminf.biomedcentral.com/articles/10.1186/1758-2946-6-10

**Nested cross-validation is recommended to estimate the prediction error (model assessment); however, we are calculating test error on three completely independent test sets. Using independent test sets for model assessment provides an unbiased estimate of test error. Krstajic et al, in the manuscript referred above by the reviewer, wrote "In an ideal situation we would have enough data to train and validate our models (training samples) and have separate data for assessing the quality of our model (test samples)". Since this is the case in this research, we are assessing the quality of our model on separate data (test sets).**
**We used LOO cross-validation for parameter tuning (model selection). Leave-one-out cross-validation is also a standard technique in the area of machine learning. Chapter 7 of *The Elements of Statistical Learning* (2009) by Hastie et al (a book cited by Krstajic et al) discusses cross-validation in detail and explains that LOO cross-validation has lower bias than 5- or 10-fold cross-validation in small datasets.**

## Validity of the findings
- In my initial review I mentioned the fact that the method is likely of limited use when applied genome-wide, because it is likely to produce a large number of false positives.
My comment was:

The accuracy of the method is likely not sufficient to be useful for applying the method on a genome-wide scale, where even a small false positive rate can lead to many erroneous predictions.

The answer of the authors was:

As we demonstrated in the revised version of the manuscript, our method outperforms or is comparable to existing approaches in terms of precision/recall and sensitivity/specificity.

While that may be true, it does not refute my initial comment. It is very unfortunate that researchers developing methods for sRNA detection choose to use relatively balanced datasets while testing their methods. This evaluation methodology gives no indication of method usability when used genome-wide (area under the precision recall curve is not invariant to the ratio of positive to negative examples), and can give biologists the wrong impression about the usability of the method. Furthermore, the majority of the features computed by the authors' method (except for the free energy feature) are insensitive to shifting the region over which they are computed. That is another reason for the limited usability genome-wide. Perhaps when combined with a method such as Barman et al, which uses primary sequence, more specificity can be obtained. Overall, it appears like the problem of sRNA identification directly from genomic sequence is a hard unsolved problem, and this should be indicated.

**We already had a sentence indicating this in the manuscript in lines 39-41: "Computational prediction of sRNAs in genomic sequences remains a challenging problem, even though tools to tackle this problem have been around since early 2000s". We have changed the wording to**

clarify this: **"Despite the fact that tools to tackle this problem have been around since early 2000s, computational prediction of sRNAs in genomic sequences remains a challenging unsolved problem"** and have added a sentence in lines 455-456 of the manuscript to indicate that there is still room for improvement in computational sRNA detection: **"Although sRNARanking outperformed the other published methods in the benchmark datasets, there is still room for improvement of computational identification of sRNAs from genomic sequences. "**.

## Comments for the Author

Minor comments:

- In line 100 n is used to denote the number of negative examples, but in lines 102-105, n is used differently to denote the relative fraction of negative to positive examples.

> **The sentences have been re-written to consistently use *n* as the number of negative examples.**

- In the description of logistic regression you reference "balanced" mode of some implementation of LR, but no mention is made which software was used. For the other classifiers specific software/version should be mentioned as well.

> **In lines 167-172 we had already mentioned that: "All the machine learning classification approaches were implemented in the Python programming language version 3.6. Scikit-learn (version 0.19.1) (Pedregosa et al., 2011) was used for the implementation of all the classifiers."**